# One-Pot Preparation of HTPB/nNi and Its Catalyst for AP

**DOI:** 10.3390/nano12152669

**Published:** 2022-08-03

**Authors:** Shuxia Bao, Tingrun Li, Chunyu Guo, Yangyang Zhao, Huijuan Zhang, Ruifeng Wu, Heping Shi

**Affiliations:** 1College of Chemical Engineering, Inner Mongolia University of Technology, Hohhot 010051, China; 2College of Science, Inner Mongolia Agricultural University, Hohhot 010018, China

**Keywords:** nano-nickel, HTPB, liquid phase reduction, one-pot method, AP, composite material

## Abstract

The liquid phase reduction method is a common method used for preparing nano-nickel powder (nNi). However, the nNi surface is easily oxidized to form nickel oxide film, which affects its performance. In this work, nNi was prepared using anhydrous ethanol as a solvent and hydrazine hydrate as a reducing agent. Furthermore, HTPB/nNi composites were prepared using hydroxyl-terminated polybutadiene (HTPB) as a coating agent. The structure and morphology of the samples are characterized by Fourier transform infrared spectroscopy (FT-IR), X-ray powder diffraction (XRD), scanning electron microscopy (SEM) and energy dispersive spectroscopy (EDS). The catalytic behavior of HTPB/nNi on the thermal decomposition of ammonium perchlorate (AP) is studied by differential scanning calorimetry (DSC) and thermogravimetric analyzer (TG). The results show that HTPB/nNi has little effect on the low temperature thermal decomposition of AP, but the peak of high temperature thermal decomposition advances from 456 °C to 332 °C.

## 1. Introduction

Nano-nickel (nNi) has applications in magnetic materials [1,2], catalysts and military [3,4] industries due to its small size effect, surface and interface effect, quantum size effect [5,6,7] and other characters. nNi has catalytic effect on propellant components and can be used to adjust the combustion performance of propellants [8,9]. Studies have shown that the heat release of the solid rocket fuel propellant is significantly increased, and its combustion performance is improved when about 1% nNi is added [10]. In addition, as a combustion accelerant of solid propellants, nNi is more effective than micron-sized nickel particles in improving the combustion rate and reducing the pressure index. Therefore, nano-nickel is an attractive high-energy material component [11,12].

Generally, the preparation methods of nNi include ionic method, spark discharge corrosion method, ball milling method and liquid phase reduction method. Bai et al. [13] obtained nNi with high purity and small average particle size by ionic method. However, this method requires special production equipment, which is difficult for industrial production. The nNi, with good sphericity and uniform particle size distribution, was prepared by the spark discharge corrosion method [14]. However, the particle size of nNi is large, and the surface of nNi is easily oxidized. Lu et al. [15] prepared nNi by high energy ball milling. The nNi has a variety of crystal forms and good stress, but nNi has low purity, uniform particle size distribution and the powder is easy to agglomerate.

The liquid phase reduction method is usually used to fabricate various metal nanoparticles. It has the advantages of a wide source of raw materials, low cost, controllable shape and size and uniform particle size distribution [16]. Hollow flower-like nickel particles were prepared by the liquid phase reduction method and used as combustion promoter in solid propellant. Importantly, the hollow spheres’ diameter can be accurately controlled from 200 nm to 2 μm by adjusting the concentration of complex Ni[C_2_H_4_(NH_2_)_2_]_3_(COOH)_2_ in the reaction solution. When a certain amount (typically 5 wt%) of the prepared nickel powders was added, the exothermic decomposition of ammonium perchlorate, which is the main component of solid propellant, was well promoted, and the heat released during decomposition increased by about 150%. In addition, the decomposition temperature was reduced by about 140 °C, and the exothermic peak at the higher temperature shifted to a lower temperature range [16].

The application of metallized propellants in solid rockets has become a common phenomenon. The combustion efficiency is favored by the activity and melting points of the metals [17]. Nano-metallic powder is easy to agglomerate due to its small particle size, large specific surface area and high surface activity, which hinders its dispersion and homogenization in the system [18,19,20]. Surfaces of nano-powders are often covered to protect them from oxidation. Hydroxyl-terminated polybutadiene (HTPB) is a suitable material to coat nNi when solid propellant applications are targeted, since it is a common binder in composite formulation for solid rocket propulsion.

In recent years, HTPB is also used for wrapping nano-metal powders [21,22]. Ju et al. [23] obtained HTPB/nAl composites by the solvent evaporation method, and studied the thermal oxidation process and safety performance at high temperature. The results show that HTPB coating improves the activity of nAl, which is beneficial to the combustion heat release, transportation and storage of nAl. Hao et al. [24] studied the HTPB-TDI coating of nano-aluminum powder in the composite spray granulation process. The results show that the coating agent can cover the surface of nano-aluminum powder, prevent the reaction of external oxygen and nano-aluminum powder and realize the long-term storage of high activity nano-aluminum powder.

In this work, an easy-to-operate liquid phase reduction method was used to reduce nickel acetate with hydrazine hydrate as a reducing agent without using surfactants, and nNi with small size and high purity was obtained. In order to prevent the oxidation of nNi, the propellant component HTPB was introduced by a one-pot method, and the nNi was directly coated to obtain the HTPB/nNi composite material. This method has the advantages of simple operation, high yield and time saving.

## 2. Experiment

### 2.1. Reagents and Instruments

Nickel acetate, anhydrous ether and sodium hydroxide (Analytical Reagent, AR) were purchased from Shanghai Mc Lin Biotech Co., Ltd. (Shanghai, China); anhydrous ethanol (AR) was purchased from Tianjin Xin ping Chemical Co., Ltd. (Tianjin, China); hydrazine hydrate (Analytical Reagent, AR, 80%) was purchased from Deheng International Trade Co., Ltd. (Heze city, China); and hydroxyl-terminated polybutadiene (AR, 80%) was purchased from Zibo Qi long Chemical Co., Ltd. (Zibo city, Shandong province, China). All the chemical reagents were used directly without any further purification. The IRTracer–100 Fourier transform infrared spectrometer (FT–IR, Shimadzu Co., Ltd., tokyo, Japan) was used to analyze the coverage of the composites, and the FT-IR data were collected in the range of 4000–400 cm^−1^. A SmartLab 9KW X-ray diffractometer (XRD, Rigaku Instrument Co., Ltd., Japan) was used to characterize the crystal structure of the sample with a scanning range of 10–90° and a scanning speed of 5 (°)/min. The XPS An ESCALAB 250 Xi multifunctional surface analysis system (XPS, Thermo) American Semerfeld Technology Ltd. The FEI company F20S–TWIN SEM transmission electron microscope (SEM) was employed to observe the surface morphology. The reactivity and stability of the powder at various temperatures and heating rates were studied with the use of the HSC-1 heat flux differential scanning calorimeter (DSC, Heng jiu Instrument Co., Ltd., Beijing, China) and the HSC–4 thermogravimetric analyzer (TG, Heng jiu Instrument Co., Ltd., Beijing, China). The samples of 5–7 mg were heated from ambient temperature to 973 K at heating rate 10 °C/min in air atmosphere. The oxidation reaction equation of nickel powder is as follows:
2Ni + O_2_→2NiO

According to the equation, the content of active nickel can be calculated [25], see Equation (1).
(1)$$m_{Ni}=2\times \frac{M_{Ni}}{M_{O_{2}}}m_{O_{2}}=3.6m_{O_{2}}$$

### 2.2. Preparation of nNi and HTPB/nNi

In order to compare the morphology and properties of nNi before and after coating, nNi without coating HTPB was prepared by the liquid phase reduction method (Figure 1).

In total, 1.5006 g (0.0085 mol) of Ni(CH_3_COOH)_2_·5H_2_O was added to 20 mL of anhydrous ethanol, and dissolved at 75 °C. The pH value of the solution was adjusted to 9 with NaOH. Finally, 13 mL of hydrazine hydrate was added to the system for reduction, keeping the temperature at 75 °C. After 30 min, the system temperature was cooled to 25 °C, and the product was washed three times with anhydrous ethanol and dried at 80 °C for 10 h. nNi (90.6%) was obtained.

In total, 0.05 g of HTPB was dissolved in 10 mL of anhydrous ether, which was then added to the above system. After continuous stirring for 2 h, the product was washed three times with anhydrous ethanol and dried at 80 °C for 10 h. HTPB/nNi (92.7%) was obtained.

### 2.3. The Catalytic Effect of HTPB/nNi on Thermal Decomposition of AP

In total, 5% HTPB/nNi was added to AP according to the previous literature [26], stirred and ground in agate mortar. An appropriate amount of anhydrous ethanol was added to the composite powder, which can make the composite powder evenly mixed. The nNi/AP and HTPB/nNi/AP composite powders were obtained after drying at 50 °C for 3 h.

## 3. Results and Discussion

### 3.1. Identification of Nickel Powder

The HTPB/nNi composites were ultrasonically dispersed in absolute ethanol. When a magnet was placed on the right side of the solution (Figure 2a), it can be observed that the composite powders aggregated toward the magnet and the powders move with the motion of the magnet (Figure 2b). Finally, most of the samples were attracted by magnet (Figure 2c). It is preliminarily proved that the Ni^2+^ is reduced to nNi. Additionally, nNi coated with HTPB can still move with the magnet, indicating that the coating does not change the valence state of nickel.

### 3.2. Characterization of X-ray Powder Diffraction (XRD)

To further confirm the successful synthesis of nNi, the XRD analysis was performed. The XRD patterns of nNi and HTPB/nNi are presented in Figure 3. The diffraction peaks at 2θ = 44.5°, 51.8° and 76.4° correspond to the (111), (200) and (220) crystal planes of face-centered cubic (fcc)-structured nickel, which are consistent with the standard phase card of Ni(ICCD: 04-0850). There is no impurity peak in the XRD patterns of nNi and HTPB/nNi, indicating that Ni^2+^ is totally reduced to nNi. An insignificant amorphous diffraction peak appears at about 2θ = 20.0°, corresponding to the coating agent HTPB, which can also be confirmed by the infrared spectrum of HTPB/nNi.

### 3.3. FT-IR Characterization

In order to determine whether HTPB was coated on the surface of nNi, the FT–IR spectra of HTPB and HTPB/nNi samples were measured and are shown in Figure 4.

The infrared absorption spectrum of HTPB is shown in Figure 4a, where 3460 cm^−1^ is the stretching vibration absorption peak of –OH, 2919 cm^−1^ is the antisymmetric stretching vibration absorption peak of C–H bond in –CH_2_, 1650 cm^−1^ is the stretching vibration absorption peak of –C=C–bond, 1437 cm^−1^ and 997 cm^−1^ are the bending vibration absorption peak and swing vibration absorption peak of –OH, 907 cm^−1^ is the in-plane bending vibration absorption peak of –CH_2_ and 715 cm^−1^ is the swing vibration absorption peak of –(CH_2_)_n_ group. The infrared spectra of HTPB/nNi (Figure 4b) show the characteristic absorption peaks of HTPB at 3736 cm^−1^ and 1532 cm^−1^, respectively, indicating that HTPB still retains its skeleton structure when it is coated on the nickel surface. However, the peak positions are shifted, which is presumed to be caused by the induction effect of nNi on –OH. In addition, the absorption peak of 997 cm^−1^ shifts to 975 cm^−1^, which is speculated to be caused by the conjugation effect of carbon–carbon double bonds. The in-plane bending vibration absorption peak of -CH_2_ (907 cm^−1^) and the swing vibration absorption peak of –(CH_2_)_n_ group (715 cm^−1^) disappeared, which may be because HTPB is regularly arranged on the surface of nNi, and the swing vibration of the group is affected by the spatial effect [27]. However, a new absorption peak appears at 661 cm^−1^, which may be the absorption peak of the Ni-O bond [28].

### 3.4. X-ray Diffraction Characterization

The XPS spectra of HTPB/nNi are shown in Figure 5. The high-resolution spectra of the 2p peak of Ni are shown in Figure 5b. The binding energies of Ni 2p_3/2_ and Ni 2p_1/2_ are relevant to pure nNi. Meanwhile, studies have shown that the Ni2p peak has two peaks of 2p_3/2_ and 2p_1/2_ [29], and there are corresponding satellite peaks near each main peak Figure 5b. The 2p_3/2_ peak at 855 eV is Ni, and their satellite peaks (861, 874) indicate that Ni^2+^ exists in the blanket. This shows that Ni in HTPB/Ni mainly exists in the form of Ni^0^, but the surface of Ni has been oxidized.

### 3.5. Morphology Characterization

The morphologies of nNi and HTPB/nNi were analyzed using SEM, as shown in Figure 5. The shape of the nNi is irregular spherical (Figure 6a) and nNi particles have a marked clustering tendency. The particles gather together into chains, resulting in the local agglomeration of nickel powder, which may be the result of the combined action of static magnetic force and surface tension between ultrafine particles [30].

It can be found from Figure 6b that HTPB is seen to form a continuous organic layer, partially adsorbed on the surface of the nanoparticles or particle clusters, and the reduced nickel powder is basically wrapped by HTPB.

The content of nickel metal in both samples is about 93 wt.% (Table 1). It can also be found that the oxygen content in HTPB/nNi (5.93%), which also included the oxygen content in HTPB, was still lower than that in Ni (6.69%), indicating that coating can effectively prevent the oxidation of nickel.

### 3.6. Thermal Performance Analysis of nNi and HTPB/nNi

The TG-DSC curve of nNi occurs in the 100 °C–700 °C ranges, as shown in Figure 7a. It can be seen that there is a distinct exothermic process in the range of 350 °C–500 °C with the peak at 387 °C from the DSC curve, and the exothermic enthalpy is 2323 J⋅g^−1^. The TG curve of nNi reveals that there is a weight gain of 12.2% in the range of 350 °C–500 °C, corresponding to the oxidation exothermic process of nNi.

According to Formula (1), about 43.92% of nNi is oxidized. In addition, there are two insignificant exothermic peaks in the temperature range of 200–350 °C, corresponding to a weight loss of about 4%, which may be due to the decomposition of a small amount of organic matter adsorbed on the surface of nNi.

The DSC curve of HTPB/nNi also has two exothermic peaks in the temperature range of 200–350 °C (Figure 7b), corresponding to a weight loss of about 8.1%, which is caused by the decomposition of HTPB [31]. In the temperature range of 350–500 °C, the oxidation exothermic process of nNi began with the peak at 382 °C, and the exothermic enthalpy is 3819 J⋅g^−1^, which is higher than that of uncoated nNi. The TG curve of HTPB/nNi has a weight gain in the range of 350–500 °C, which is corresponding to the oxidation exothermic process of *n*Ni, and the weight gain is about 18.7%. According to Formula (1), about 67.32% of nNi is oxidized, which is higher than that of pure nNi (43.92%). The TG–DSC analysis results of nNi and HTPB/nNi indicate that HTPB coating on the surface of nNi can prevent the surface oxidation of nNi and the content of active nickel was increased.

### 3.7. Analysis of HTPB/nNi on the Thermal Decomposition Performance of AP

AP is a commonly used oxidant in propellants, accounting for 60–90% of solid rocket propellants. Their thermal decomposition characteristics have a great influence on the burning rate of propellants [32,33]. In order to improve the combustion performance of propellants, a common alternative way is to utilize novel energetic additives [34]. Energetic combustion catalyst can not only reduce the thermal decomposition temperature, but also increase the combustion heat release of propellants [35]. Ni nanoparticles exhibit higher catalytic activity on the thermal decomposition of AP due to their smaller size and larger specific surface [10]. Unfortunately, unprotected Ni nanoparticles are susceptible to irreversible aggregation in the solution due to their small size and magnetism, which will lower their catalytic activity. One effective strategy is to coat the Ni surface, which can prevent aggregation and maintain its catalytic activity [36]. The combustion performance of a propellant can be speculated by studying the thermal decomposition characteristics of AP in a propellant with metal catalysts [37]. In this work, TG–DSC (heating rate of 10 °C·min^−1^) was used to analyze the effect of HTPB/nNi on the thermal performance of AP, and the curves are shown in Figure 8.

Figure 8a is the DSC curve of AP, nNi/AP and HTPB/nNi/AP, respectively. It can be seen that the pure AP has one endothermic peak and two exothermic peaks in the temperature range of 80–480 °C. The endothermic peak at 221 °C is a crystal transition process, and the AP transforms from tetragonal to cubic. The exothermic peak at 280–330 °C is a low-temperature decomposition process, and the peak is 315 °C. The exothermic process at 400–470 °C is a high-temperature decomposition process with the peak of 456 °C, and the AP is completely decomposed [38].

When nNi and HTPB/nNi (both 5%) were added, the decomposition process of AP was affected. Compared with pure AP, nNi and HTPB/nNi delayed the endothermic peak of AP by 17 °C and 14 °C, respectively. The low–temperature decomposition peaks of AP were advanced by 15 °C. Generally, reducing the thermal decomposition temperature of AP can promote the full combustion of solid propellants at lower temperatures [39]. nNi and HTPB/nNi advance the high-temperature decomposition peak (456 °C) of AP to 356 °C and 332 °C, respectively. The exothermic peaks of high-temperature decomposition were more advanced, indicating that the catalytic effect of nNi and HTPB/nNi on the high-temperature thermal decomposition reaction of AP was more obvious than that on the low-temperature thermal decomposition reaction of AP. Among them, the catalytic effect of HTPB/nNi on the high-temperature decomposition reaction of AP is more prominent, which plays a synergistic effect of composite materials and shows better catalytic effect.

It can be seen from Figure 8b that the initial thermal decomposition temperatures of AP, nNi/AP and HTPB/nNi/AP are 285 °C, 273 °C and 268 °C, respectively. The termination temperatures of thermal decomposition were 424 °C, 380 °C and 377 °C, respectively. The temperature differences were 139 °C, 107 °C and 109 °C, respectively, which means that the thermal decomposition rate of AP is gradually accelerated. In this temperature range, the samples of AP, nNi/AP and HTPB/nNi/AP all lost weight by 100% and can be judged to be completely decomposed of AP, which is consistent with the conclusion in the literature [40].

## 4. Conclusions

Without introducing non-propellant components, HTPB/nNi was prepared by reduction and coating in one pot. As a coating agent, HTPB can protect nNi from oxidation and the introduction of HTPB does not change the crystal form of nickel. The active nickel content of the coated nNi increased from 45.0% to 69.0%.The application of nano-nickel was greatly improved after coating. The HTPB/nNi composite can reduce the low-temperature decomposition peak (315 °C) and high-temperature decomposition peak (456 °C) of AP to 300 °C and 332 °C, respectively, which improves the thermal decomposition efficiency of AP.

## Figures and Tables

**Figure 1 nanomaterials-12-02669-f001:**
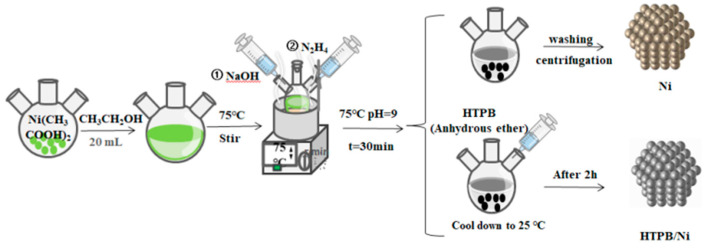
Flow chart of preparation of nNi and HTPB/Ni composite particles.

**Figure 2 nanomaterials-12-02669-f002:**
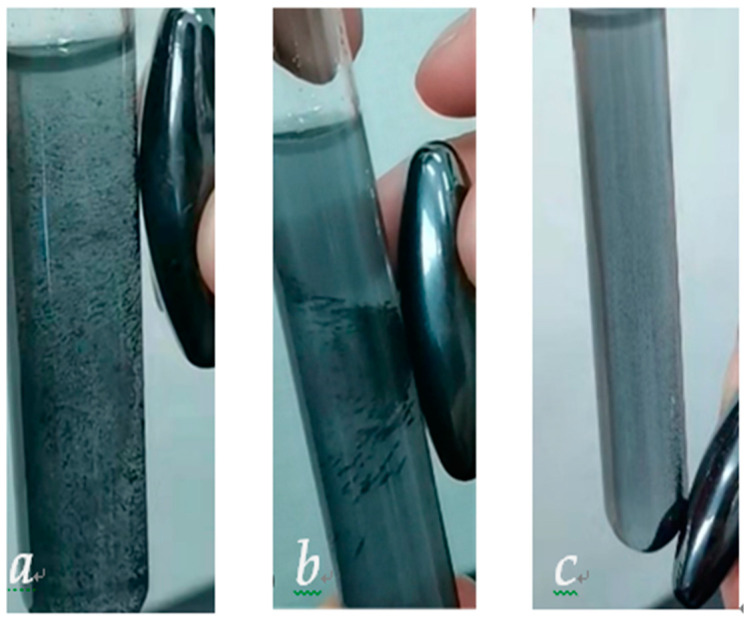
HTPB/nNi dispersed in anhydrous ethanol ((**a**) Before attraction; (**b**) During attraction; (**c**) After attraction).

**Figure 3 nanomaterials-12-02669-f003:**
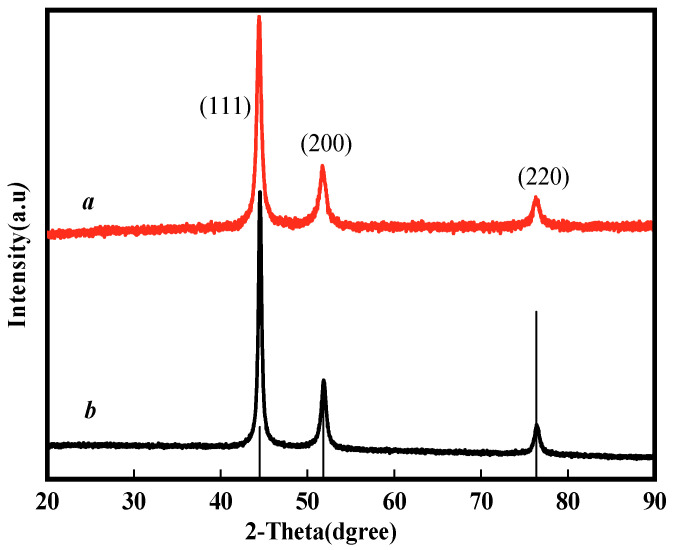
XRD patterns of HTPB/nNi ((**a**) HTPB/nNi; (**b**) nNi).

**Figure 4 nanomaterials-12-02669-f004:**
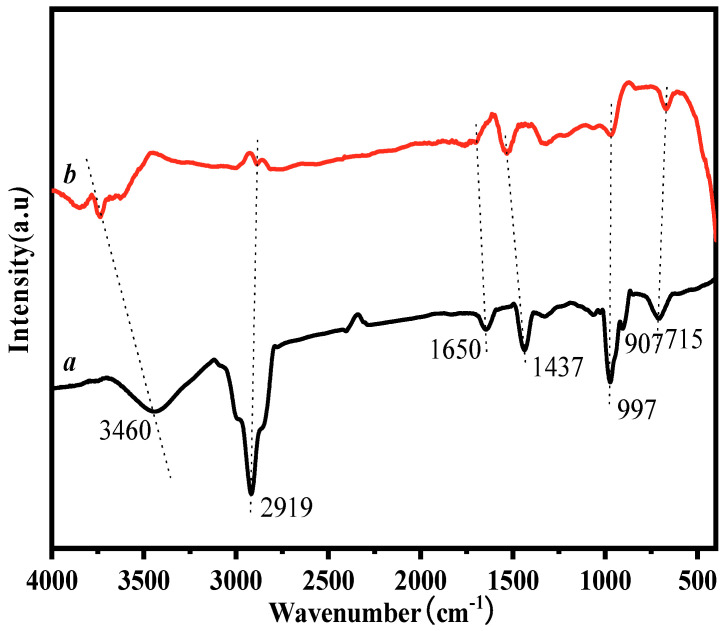
FT−IR spectra of HTPB (**a**) and HTPB/nNi (**b**).

**Figure 5 nanomaterials-12-02669-f005:**
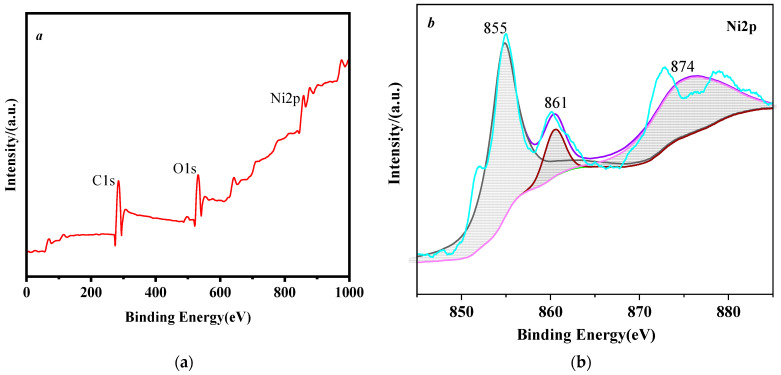
XPS spectra scanning of HTPB/Ni. (**a**) Full–spectrum scanning and (**b**) 2p peak of Ni.

**Figure 6 nanomaterials-12-02669-f006:**
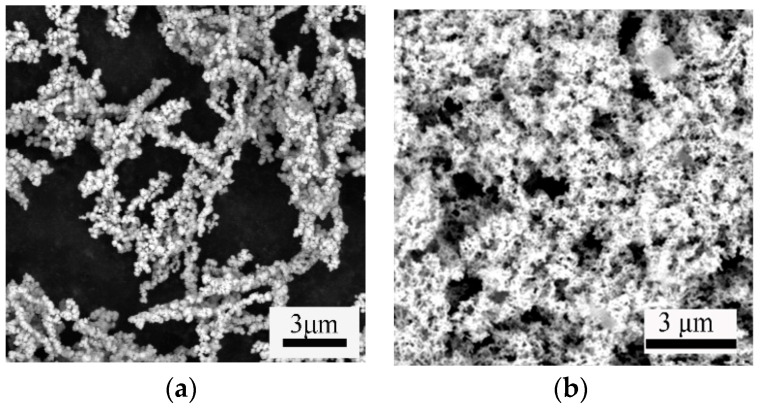
SEM images of nNi (**a**) and HTPB/nNi (**b**).

**Figure 7 nanomaterials-12-02669-f007:**
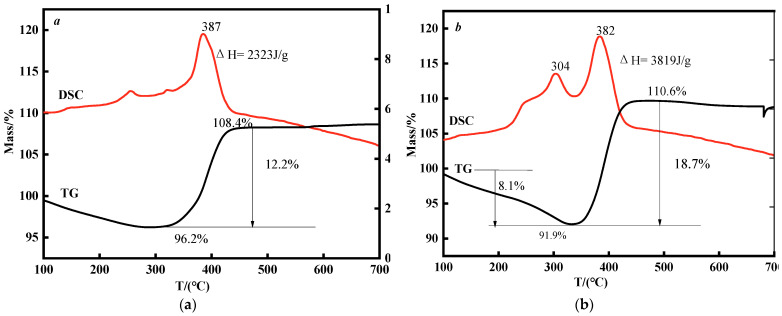
TG-DSC curves of nNi (**a**) and HTPB/nNi (**b**).

**Figure 8 nanomaterials-12-02669-f008:**
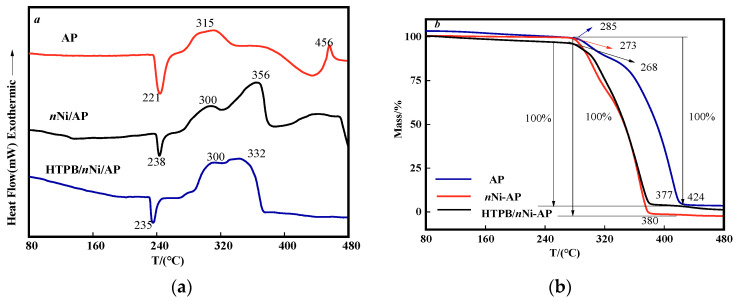
DSC (**a**) and TG (**b**) curves of AP.

**Table 1 nanomaterials-12-02669-t001:** EDS analysis results of nNi and HTPB/nNi.

Sample	Ni	O	C	Total Amount
nNi	93.31	6.69	—	100.00
HTPB/nNi	93.48	5.93	0.59	100.00

## Data Availability

Exclude this this statement because the study did not report any data.

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
