# Peer review of "One-Pot Preparation of HTPB/nNi and Its Catalyst for AP"

_nanomaterials, 2022, doi:10.3390/nano12152669_

Round 1

Reviewer 1 Report

The description of the thermal analysis should be given in the experimental part. 

1.  In air atmosphere, the thermal behaviors of nNi and HTPB/nNi in the temperature range of 100 °C700 °C were investigated at a heating rate of 10 °C·min-1.

2. And According to the equation, the content of active nickel can be calculated[30], see 195 formula (1). 196 ??? = 2 × ??? ??2 ??2 = 3.6??2 (1)

3. Conclusion must be improved

4. Can your method be used for Ni alloys?

Author Response

Dear professor,

We are very grateful for giving us the chance to submit the revised manuscript, and we would like to thank the reviewers for giving us constructive suggestions that not only helped us in English but also in depth improved the quality of manuscript. Here we submit a new version of our manuscript with the title “One-pot preparation of HTPB/nNi and its catalytic for AP” (nanomaterials-1807371), which has been modified according the reviewers’ suggestions. We mark all the changes by “Track Changes” in the revised manuscript.

The following is a point-to-point response to the reviewers’ comments.

Reviewers' comments:

Reviewer #1:

Comment (1): The description of the thermal analysis should be given in the experimental part.

  1. In air atmosphere, the thermal behaviors of nNi and HTPB/nNi in the temperature range of 100 °C-700 °C were investigated at a heating rate of 10 °C·min-1.
  2. And According to the equation, the content of active nickel can be calculated [30], see 195 formula (1). 196 ??? = 2 × ??? ??2 ??2 = 3.6??2 (1)
  3. Conclusion must be improved
  4. Can your method be used for Ni alloys?

Response: Thank you for your valuable comments on the manuscript. For your comments, make the following reply.

  1. The description of thermal analysis has been placed in the experimental part.(For more information, see lines 101-105)
  2. Conclusions have been improved. (For more details, line 291-293)

I think the alloy needs two or more metals to be melted at high temperature and cooled. The reduction process of nickel was carried out at 75 °C.

Reviewer 2 Report

Nanomaterials

Manuscript ID:           Nanomaterials-1807371

Title:                            One-pot preparation of HTPB/nNi and its catalytic for AP

In this manuscript, the authors prepared nano-nickel powder by anhydrous ethanol as solvent. Furthermore, HTPB/nNi composites were prepared using hydroxyl-terminated polybutadiene (HTPB) as a coating agent. The physicochemical properties were investigated using FT-IR, XRD, SEM and EDS. I think that is an interesting work, and I recommend publication to Nanomaterials journal after the authors consider the following major revisions.

Comment #1

It would be very useful for the authors to take measurements via X-ray photoelectron spectroscopy for all samples in order to investigate surface coverage or the nickel species.

Comment #2

How is the catalytic stability of these samples determined?

 Comment #3

Apart from XRD; I think that the authors should utilize TEM analysis on the samples in order to investigate the Ni particle size.

Comment #4

Is it possible for the authors to calculate the Ni0 metal particle size and dispersion for the catalysts using TEM analysis? Is there any structural change?

Author Response

Dear professor,

We are very grateful for giving us the chance to submit the revised manuscript, and we would like to thank the reviewers for giving us constructive suggestions that not only helped us in English but also in depth improved the quality of manuscript. Here we submit a new version of our manuscript with the title “One-pot preparation of HTPB/nNi and its catalytic for AP” (nanomaterials-1807371), which has been modified according the reviewers’ suggestions. We mark all the changes by “Track Changes” in the revised manuscript.

Reviewer:

Title: One-pot preparation of HTPB/nNi and its catalytic for AP

In this manuscript, the authors prepared nano-nickel powder by anhydrous ethanol as solvent. Furthermore, HTPB/nNi composites were prepared using hydroxyl-terminated polybutadiene (HTPB) as a coating agent. The physicochemical properties were investigated using FT-IR, XRD, SEM and EDS. I think that is an interesting work, and I recommend publication to Nanomaterials journal after the authors consider the following major revisions.

Comment #1: It would be very useful for the authors to take measurements via X-ray photoelectron spectroscopy for all samples in order to investigate surface coverage or the nickel species.

Comment #2: How is the catalytic stability of these samples determined?

Comment #3: Apart from XRD; I think that the authors should utilize TEM analysis on the samples in order to investigate the Ni particle size.

Comment #4: Is it possible for the authors to calculate the Ni0 metal particle size and dispersion for the catalysts using TEM analysis? Is there any structural change?

Response: Thank you for your valuable comments on the manuscript. For your comments, make the following reply.

Comment #1: The samples have been measured by X-ray photoelectron spectroscopy (For details, see lines 175-189).

Comment #2: The stability of the thermal catalyst can be measured by thermal stability. The curves of DSC and TG can prove the stability of the sample.

Comment # 3 and 4: Because the magnetism of the sample will interfere with the normal operation of the electron microscope, the sample cannot be photographed clearly. Scanning electron microscopy showed that the reduced nickel was irregular spherical. We will try to find a TEM test institution willing to provide test of magnetic sample in the further work.

Round 2

Reviewer 2 Report

-